OBSERVATION

# Ejaculate for Microbiological Culture: To Wash or Not To Wash?

Tom Theiler,[a] ⓘ Ioana Diana Olaru,[a] Charlotte Kilzer,[a] Franziska Schuler,[a] Frieder Schaumburg[a]

[a]Institute of Medical Microbiology, University Hospital Münster, Münster, Germany

**ABSTRACT** Bacteria can be associated with male infertility. Antibacterial substances (e.g., zinc-containing proteins, antimicrobial peptides) in ejaculates might impair the growth of bacteria in culture. We therefore wanted to test if removing antibacterial substances by washing the ejaculate could improve the detection of bacteria in culture. All ejaculates from patients ≥18 years old, which were obtained for routine diagnostics to assess male infertility were included in this study (no exclusion criteria were applied). Test samples were diluted with 2 mL sterile 0.45% saline, vortexed, and centrifuged (5 min; $7.5 \times g$). After the removal of 2 mL of the supernatant and resuspension, 10 $\mu$L of the pellet was used for aerobic and anaerobic culture. Control samples were cultured identically but without washing. Species identification was done with matrix-assisted laser desorption ionization–time of flight mass spectrometry. A total of 186 samples were included. The data set was stratified into five groups (Gram-negative rods [GNR], anaerobes [AN], *Enterococcus* spp. [EC], coagulase-negative staphylococci [CNS], and viridans streptococci [VS]). Compared to the control arm, the test arm revealed significant lower proportions for CNS (59.1% versus 44.6%, $P < 0.01$) and VS (53.8% versus 41.9%, $P = 0.03$). Similarly, slightly lower proportions of GNR (16.1% versus 15.1%, $P = 0.89$), AN (19.9% versus 17.2%, $P = 0.5$), and EC (25.3% versus 23.1%, $P = 0.63$) were observed. The medians of CFU were lower in test samples compared to the control samples ($6.5 \times 10^3$ versus $2.5 \times 10^3$, $P < 0.01$) for any bacterial growth. Lower colony counts were also observed for individual bacterial groups. In conclusion, preculture washing of ejaculates results in a decrease in total bacteria count and culture-positive samples.

**IMPORTANCE** This study compares two methods for processing ejaculate samples from men undergoing investigations for infertility. The method of sample washing and centrifugation was compared to the standard method of direct inoculation and culture. The study hypothesis was that preprocessing of samples may increase bacterial yield by removing bactericidal substances from semen. However, we found that washing ejaculate samples before microbiological culture did not improve the detection of bacteria and led to a reduction in colony counts.

**KEYWORDS** semen, male infertility, bacteria, method evaluation, test evaluation

**M**ale infertility affects nearly 7% of all men, and its etiology remains unclear in around 50% of cases (1). It is estimated that 15% of male infertility is caused by genital tract infections (2). However, the role of bacteria in male infertility remains poorly understood. A challenging factor in microbiological diagnostics of ejaculate is a high level of contamination from the perianal region (3). In addition, the microbiome of ejaculates is diverse both in infertile and fertile men and only a small proportion of the species is truly associated with infertility such as Gram-negative and anaerobic bacteria (4). One of the most relevant bacteria in male infertility is *Escherichia coli*: *in vitro* studies have shown a direct toxic effect on spermatozoa by affecting the mitochondrial membrane potential (5). Patients with asthenospermia and oligoasthenospermia had a larger amount of Gram-negative bacteria in their ejaculate that contained lipopolysaccharide (LPS) in the cell wall. LPS can cause inflammation by upregulating cytokines (6).

Address correspondence to Ioana Diana Olaru, ioanadiana.olaru@ukmuenster.de.

The authors declare no conflict of interest.

**TABLE 1** Comparison of samples with or without prewashing for the 186 samples included in the study[a]

| Bacterial group | No washing (control arm) [N (%)] | Washing prior culture (test arm) [N (%)] | P value[c] | No washing (control arm) [median CFU (×10³/mL IQR)] | Washing prior culture (test arm) [median CFU (×10³/mL IQR)] | P value[d] |
|---|---|---|---|---|---|---|
| Culture-positive samples | 178 (95.7) | 158 (84.9) | <0.01 | 6.5 (2.3–32.0) | 2.5 (0.2–12.0) | <0.01 |
| Gram-negative rods | 30 (16.1) | 28 (15.1) | 0.89 | 3.2 (0.4–14.9) | 1.0 (0.2–3.7) | 0.19 |
| Anaerobes[b] | 37 (19.9) | 32 (17.2) | 0.50 | 4.2 (1.2–19.6) | 3.0 (0.3–26.5) | 0.71 |
| Enterococcus spp. | 47 (25.3) | 43 (23.1) | 0.63 | 3.5 (1.3–22.1) | 2.6 (0.5–7.8) | 0.12 |
| Coagulase-negative staphylococci | 110 (59.1) | 83 (44.6) | <0.01 | 0.8 (0.2–2.5) | 0.5 (0.2–1.8) | 0.22 |
| Viridans streptococci | 100 (53.8) | 78 (41.9) | 0.03 | 2.7 (0.6–6.4) | 1.1 (0.3–7.4) | <0.01 |

[a]Only species/genera were considered if ≥30 isolates were tested in each group. IQR, interquartile range.
[b]Anaerobes identified belonged to *Peptoniphilus* spp., *Actinobaculum* spp., *Actinomyces* spp., *Alloscardovia* spp., *Anaerococcus* spp., *Bacteroides* spp., *Cutibacterium* spp., *Finegoldia* spp., *Fusobacterium* spp., *Granulicatella* spp., *Peptostreptococcus* spp., *Prevotella* spp., and *Veillonella* spp.
[c]P value from chi-square test, comparison of proportions.
[d]P value from Mann-Whitney U test, comparison of CFU counts.

Antibacterial substances in the ejaculate hamper the detection of bacteria by microbiological culture, particularly *Neisseria gonorrhoeae* (7). For instance, Zn-containing proteins maintain sperm membrane stability leading to reduced sperm mobility, which can lead to a lower probability of fertilization of the ovum (8). Antimicrobial peptides are able to interact with cells of the innate immune system, such as neutrophils, monocytes, macrophages, and epithelial cells, or lead to cytolysis due to insertion of pores in the bacterial cell membrane (8, 9). Thus, the removal or dilution of bactericidal substances from ejaculate could improve the sensitivity of microbiological culture. Therefore, washing the ejaculate before culture was suggested (10, 11). However, there is only evidence that preculture dilution with sodium chloride improves the detection of *N. gonorrhoeae* (7). It is unclear if this is also the case for other, more common bacterial species that are associated with male infertility. Therefore, the aim of this study was to compare the detection of bacteria from ejaculate samples with or without washing before culture.

In total, 186 samples comprising 588 different bacterial isolates were included. *Lactobacillus* sp. ($n = 12$, control arm; $n = 10$, test arm), *Corynebacterium* sp. ($n = 10$, control arm; $n = 7$, test arm), beta-hemolytic streptococci ($n = 9$, control arm; $n = 5$, test arm) were not included due to sample sizes <30.

The proportion of samples, which were positive for any of the defined groups (i.e., Gram-negative rods [GNR], anaerobes [AN], Enterococcus spp. [EC], coagulase-negative staphylococci [CNS], and viridans streptococci [VS]) was 95.7% ($n = 178/186$, control arm) and 84.9% ($n = 158/186$, test arm). Samples were positive for CNS (59.1%, control arm versus 44.6%, test arm), followed by viridans streptococci (53.8%, control arm versus 41.9%, test arm), *Enterococcus* spp. (25.3%, control arm versus 23.1%, test arm), and anaerobes (19.9%, control arm versus 17.2%, test arm) (Table 1). Samples without washing were more often positive for any of the target organisms compared to those being washed before culture (Fig. 1). This difference reached statistical significance in CNS (59.1% versus 44.6%, $P < 0.01$) and VS (53.8% versus 41.9%, $P = 0.03$, Table 1).

In general, the median CFU was significantly higher in the control arm (no washing) compared to the test arm (washing before culture, $6.5 \times 10^3$/mL versus $2.5 \times 10^3$/mL, $P < 0.01$, Table 1). A higher count of CFU per milliliter was also observed for GNR, AN, EC, and CNS and was significantly higher for VS.

We compared the detection of bacteria in ejaculate with and without washing before culture and found that washing both reduced the proportion of positive samples and the median CFU of detected bacteria. Before conducting this study, we had assumed that washing would remove potential bactericidal substances from ejaculate samples and it is therefore surprising that both the total count and the proportion of positive samples were lower in the test arm (washing before culture). This contradicts the general recommendation to wash ejaculate samples to increase culture sensitivity (10).

Several reasons for better detection of species without washing are possible: first, contrary to our original hypothesis, bacteria, and other growth-promoting factors may have been removed during the washing process leading to lower bacterial counts in the processed

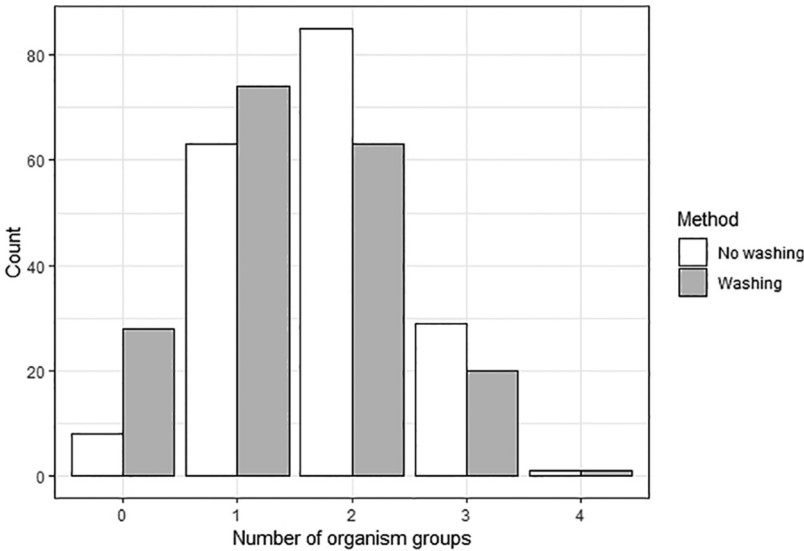

**FIG 1** Number of organism groups per sample using the two methods.

sample. Second, centrifugal forces and sheer stress during centrifugation could have impaired the viability of bacterial cells. For instance, centrifugal compaction can cause cell surface damage of *Staphylococcus* spp. (12). Finally, different sample volumes were used for washing, which may have affected bacterial yield.

Our study has limitations. First, we did not screen for *Ureaplasma* spp. and *Mycoplasma* spp., and thus, we are unable to assess the impact of washing on these genera. However, the majority of laboratories use nucleic acid amplification tests for their detection, and it is unlikely that bactericidal substances have an impact on molecular methods.

In conclusion, washing ejaculate samples before microbiological culture does not improve the detection of bacteria. Therefore, ejaculate washing could be omitted before culture.

**Ethics.** Ethical approval was obtained from our institutional review board (IRB; Ethik Kommission der Ärztekammer Westfalen-Lippe und der Westfälischen Wilhelms-Universität; 2021-802-f-S). The IRB granted a waiver for obtaining signed written informed consent from patients.

**Study design.** We performed a prospective cross-sectional study at the University Hospital Münster (UKM), Germany from September 2021 to February 2022. Inclusion criteria were as folllows: (i) patients treated at the UKM for male infertility and (ii) age ≥18 years. No exclusion criteria were applied. Approximately, 500 ejaculate samples are sent for microbiological culture per year. The endpoints of the study were (i) number of CFU for each bacterial group and (ii) proportion of individual species in ejaculate with and without washing before culture.

**Microbiological analyses.** Ejaculate samples were processed in parallel and allocated to both the control and test arm. In the control arm, culture was done without washing. In the test arm, samples were washed before culture.

In the control arm, 10 $\mu$L of ejaculate was directly streaked on MacConkey agar (Oxoid, Wesel, Germany; culture at 35 $\pm$ 2°C for 24 h, ambient air), chocolate agar (Oxoid; culture at 35 $\pm$ 2°C for 24 h under 5% $CO_2$), and Schaedler agar (Oxoid; culture at 35 $\pm$ 2°C for 48 h, anaerobic conditions) using a calibrated loop. Agar plates were inoculated using the continuous streaking method for the accurate quantification of CFU.

In the test arm, 100 to 300 $\mu$L of the ejaculate was suspended with 2 mL sterile 0.45% sodium chloride and centrifuged (7.5 $\times$ *g*, room temperature) for 5 min (3). To remove bactericidal substances, 2 mL of the supernatant was discarded. Since the same volume was added and removed after the washing, neither a dilution nor a concentration of CFU should be expected. Of the pellet, 10 $\mu$L was cultured identically as the control arm.

If bacterial growth was detected, species identification was performed by matrix-assisted laser desorption ionization–time of flight mass spectrometry (Bruker, Bremen, Germany)

using the 4.2 IVD Compass library. The bacterial concentration (in CFU/mL) was counted for each species in individual samples.

**Sample size calculation.** In the absence of appropriate assumptions for the proportion of individual species in ejaculate samples with or without washing, we were unable to perform a sample size calculation and used a convenience sample instead. Bacteria were categorized into five groups (GNR, AN, EC, CNS, and VS). Recruitment was stopped until ≥30 isolates were available for each of the groups (GNR, AN, EC, CNS, VS). The Clinical and Laboratory Standards Institute requires that a minimum of 30 isolates should be available for meaningful pathogen statistics (13).

**Statistical analysis.** Categorical variables (presence/absence of a species) were compared between the control and test arm using the Chi-square test. Continuous variables (CFU) were first tested for normal distribution using the Anderson-Darling test method. Nonnormally distributed variables were compared between the two arms using the Mann-Whitney U test. Analyses were performed with R 4.1.2 (R Foundation for Statistical Computing, Vienna, Austria). The significance level was set at 0.05.

**Data availability.** All data analyzed are available in the manuscript and Table 1.

## ACKNOWLEDGMENTS

We are grateful for the excellent support of our laboratory technicians at the Institute of Medical Microbiology.

All authors had full access to the data in the study and take responsibility for the integrity of the data and the accuracy of the data analysis.

T. Theiler contributed Conceptualization, Data Curation, Formal Analysis, Investigation, Methodology, Writing – Original Draft Preparation. Ioana-Diana Olaru contributed Formal Analysis, Data Curation, Methodology, Writing – Review & Editing; Charlotte Kilzer contributed data Curation, Writing – Review & Editing; Franziska Schuler contributed Conceptualization, Investigation, Methodology, Writing – Review & Editing. F. Schaumburg contributed Conceptualization, Investigation, Formal Analysis, Resources, Supervision, Writing – Original Draft Preparation, Writing – Review & Editing.

We acknowledge the support of the open access fund of the University of Münster. No external funding was received for the study. We declare no conflicts of interest.

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
