## [Reviewer comments · Microbiology Spectrum]

Microbiology Spectrum

Ejaculate for microbiological culture: to wash or not to wash?

Tom Theiler, Ioana Diana Olaru, Charlotte Kilzer, Franziska Schuler, and Frieder Schaumburg

Corresponding Author(s): Ioana Diana Olaru, University Hospital Münster,

Review Timeline:

Submission Date:	August 18, 2022
Editorial Decision:	September 26, 2022
Revision Received:	October 7, 2022
Accepted:	October 17, 2022

Editor: William Lainhart

Reviewer(s): The reviewers have opted to remain anonymous.

Transaction Report:

DOI: <https://doi.org/10.1128/spectrum.03269-22>

September 26, 2022

Dr. Ioana Diana Olaru
University Hospital Münster,
Institute of Medical Microbiology
Domagkstr 10
Münster 48149
Germany

Re: Spectrum03269-22 (Ejaculate for microbiological culture: to wash or not to wash?)

Dear Dr. Ioana Diana Olaru:

Link Not Available

Sincerely,

William Lainhart

Journals Department
Reviewer comments:

Reviewer #1 (Comments for the Author):

The authors describe a study in which bacterial recovery was compared in washed and unwashed ejaculate specimens. The authors conclude that there should not be a washing step. The data in Table 1 support this conclusion. The following are suggestions for strengthening the manuscript:

1. Table 1 displays the median CFU/mL of each culture, but what was the mean? Consider displaying this data in the supplement.
2. Were these the only bacterial groups that were recovered? Were there coryneform, Neisseria, or Staphylococcus aureus? What about oxidase-positive GNRs? Could there be an "other" grouping?

3. Can the types of anaerobes encountered be defined?
4. Was there more than one grouping of bacteria per sample? Perhaps create a pie chart with the number of bacterial groupings per sample.
5. In table 1, remove N=186 from the first column.
6. Consider "method evaluation" instead of "test evaluation" as a keyword.
7. Line 16: spell out Zinc the first time Zn is used
8. Line 17: suggest rephrasing to " might impair growth of bacteria in culture."
9. Line 25: MALDI-TOF should have "mass spectrometry" after it
10. Gram should be capitalized throughout
11. Line 75: Remove sentence that begins with "The center..."
12. Line 112: spell out Chi square test
13. Line 148, 149: spp. should not be italicized

Reviewer #2 (Comments for the Author):

In the manuscript by Theiler et al, the authors evaluated if washing ejaculate specimens with saline (test) prior to culture would improve recovery of bacteria relative to unwashed specimens (control). Overall, the number of positive cultures was significantly lower in the test group and lower colony counts were observed, relative to the control group.

- A. On lines 17, 19 and 140, consider changing "cultural" to "culture".
- B. In the Material and Methods, how was the 10 uL of specimen plated? Was it pipetted or was a calibrated loop used?
- C. Were the control specimens plated prior to the test specimens or were the controls sitting at room temperature? How much of a time difference and could this account for lower yields?
- D. On lines 124-125 it states "Most samples were positive for CNS, followed by viridans streptococci, Enterococcus spp and anaerobes" Please clarify what "most samples" means.

Staff Comments:

Preparing Revision Guidelines

Please return the manuscript within 60 days; if you cannot complete the modification within this time period, please contact me. If you do not wish to modify the manuscript and prefer to submit it to another journal, please notify me of your decision immediately so that the manuscript may be formally withdrawn from consideration by Microbiology Spectrum.

October 7th, 2022

Dear Editor,

Thank you for considering our manuscript for publication and for sending comments from peer reviewers. Please find below a detailed response to the editorial comments and reviewers' comments.

REVIEWER 1

Comment 1

Table 1 displays the median CFU/mL of each culture, but what was the mean? Consider displaying this data in the supplement.

Response to comment 1

Thank you for your comment. We felt that the median would provide a better representation of the distribution of the data given that data are not normally distributed. We have added a table below with mean CFUs to show that the results and interpretation would not change.

	No washing (Control arm) Mean CFU ($\times 10^3$ /ml SD)	Washing prior culture (Test arm) Mean CFU ($\times 10^3$ /ml SD)
Culture positive samples	32.8 (106.9)	28.4 (110.2)
Gram-negative rods	46.1 (176.0)	44.7 (186.0)
Anaerobes	18.3 (28.7)	31.5 (67.0)
Enterococcus spp.	15.4 (23.8)	11.9 (26.4)
Coagulase-negative staphylococci	3.5 (12.8)	4.3 (11.6)
Viridans streptococci	8.8 (19.0)	7.4 (19.7)

Comment 2

Were these the only bacterial groups that were recovered? Were there coryneform, Neisseria, or Staphylococcus aureus? What about oxidase-positive GNRs? Could there be an "other" grouping?

Response to comment 2

Only bacteria (or groups) that were present in at least 30 samples were reported separately for the purpose of this comparison. In samples which did not undergo washing, we identified 10 samples with *Corynebacterium* spp., 1 with *S. aureus*, 1 with *P. aeruginosa* and 1 with *Neisseria* spp. (other than *N. gonorrhoea*). In samples with washing, we identified 7 samples with *Corynebacterium* spp, 1 with *S. aureus*, and 1 with *P. aeruginosa*.

Comment 3

Can the types of anaerobes encountered be defined?

Response to comment 3

We identified the following anaerobes: *Peptoniphilus*, *Actinomyces*, *Peptoniphilus*, *Veillonella*, *Bacteroides*, *Granulicatella*, *Anaerococcus*, *Fusobacterium*, *Fingoldia*,

Cutibacterium, *Peptostreptococcus*, *Actinobaculum*, *Prevotella*, *Alloscardovia*. The information was added in the Table footnote.

Comment 4

Was there more than one grouping of bacteria per sample? Perhaps create a pie chart with the number of bacterial groupings per sample.

Response to comment 4

Yes, samples frequently had growth of multiple groups. A graph displaying the number of organism groups using the two methods was added to the manuscript (Figure 1).

Comment 5

In table 1, remove N=186 from the first column.

Response to comment 5

N=186 was removed from the first column.

Comment 6

Consider "method evaluation" instead of "test evaluation" as a keyword.

Response to comment 6

The change was made as advised.

Comment 7

Line 16: spell out Zinc the first time Zn is used

Response to comment 7

Changed as suggested.

Comment 8

Line 17: suggest rephrasing to "might impair growth of bacteria in culture."

Response to comment 8

The sentence was rephrased.

Comment 9

Line 25: MALDI-TOF should have "mass spectrometry" after it

Response to comment 9

Changed as suggested.

Comment 10

Gram should be capitalized throughout

Response to comment 10

The changes were made as advised.

Comment 11

Line 75: Remove sentence that begins with "The center..."

Response to comment 11

The sentence was removed.

Comment 12

Line 112: spell out Chi square test

Response to comment 12

Changed as suggested.

Comment 13

Line 148, 149: spp. should not be italicized

Response to comment 13

The changes were made as advised.

REVIEWER 2**Comment 1**

On lines 17, 19 and 140, consider changing "cultural" to "culture".

Response to comment 1

The sentence was rephrased to "might impair growth of bacteria in culture."

Comment 2

In the Material and Methods, how was the 10 uL of specimen plated? Was it pipetted or was a calibrated loop used?

Response to comment 2

A 10 µL calibrated loop was used. The information was added in the methods section.

Comment 3

Were the control specimens plated prior to the test specimens or were the controls sitting at room temperature? How much of a time difference and could this account for lower yields?

Response to comment 3

The test plates were inoculated immediately after the control plates. In rare cases this was not possible and the sample was cooled to 2-8 degrees until plating. The washing process usually lasted for less than 10 minutes and therefore this is unlikely to have contributed to the lower yield.

Comment 4

On lines 124-125 it states "Most samples were positive for CNS, followed by viridans streptococci, Enterococcus spp. and anaerobes" Please clarify what "most samples" means.

Response to comment 4

The sentence was clarified: "*Samples were positive for CNS (59.1%, control arm vs. 44.6%, test arm), followed by viridans streptococci (53.8%, control arm vs. 41.9%, test arm), Enterococcus spp. (25.3%, control arm vs. 23.1%, test arm) and anaerobes (19.9%, control arm vs. 17.2%, test arm) (Table 1).*"

We are very grateful to the reviewers for the feedback provided for improving this manuscript. In the hope that we have responded to the comments satisfactorily,

Yours sincerely,

Ioana D Olaru

October 17, 2022

Dr. Ioana Diana Olaru
University Hospital Münster,
Institute of Medical Microbiology
Domagkstr 10
Münster 48149
Germany

Re: Spectrum03269-22R1 (Ejaculate for microbiological culture: to wash or not to wash?)

Dear Dr. Ioana Diana Olaru:

Your manuscript has been accepted, and I am forwarding it to the ASM Journals Department for publication. You will be notified when your proofs are ready to be viewed.

Sincerely,

William Lainhart
Editor, Microbiology Spectrum
